# Impacts of an Intense Wildfire Smoke Episode on Surface Radiation, Energy and Carbon Fluxes in Southwestern British Columbia, Canada

I.G. McKendry[1], A. Christen[2], S.-C. Lee[1], M. Ferrara[1], K.B. Strawbridge[3], N. O'Neill[4], A. Black[5]

[1] Department of Geography, The University of British Columbia, Vancouver, Canada

[2] Environmental Meteorology, Faculty of Environment and Natural Resources, University of Freiburg,
Freiburg, D-79085, Germany

[3] Air Quality Research Division, Environment Canada, 4905 Dufferin St, Toronto, Canada

[4] Centre for Research and Applications in Remote Sensing, Université de Sherbrooke, Sherbrooke, Canada

[5] Faculty of Land and Food Systems, The University of British Columbia, Vancouver, Canada

*Correspondence to*: Ian McKendry (ian@geog.ubc.ca)

## Abstract

A short, but severe, wildfire smoke episode in July 2015, with an aerosol optical depth
(AOD) approaching nine, is shown to strongly impact radiation budgets across four
distinct land use types (forest, field, urban and wetland). At three of the sites, impacts on
the energy balance are also apparent, while the event also appears to elicit an ecosystem
response with respect to carbon fluxes at the bog and a forested site. Greatest impacts on
radiation and energy budgets were observed at the forested site where the role of canopy
architecture, and the complex physiological responses to an increase in diffuse radiation
were most important. At the forest site, the arrival of smoke reduced both sensible and
latent heat flux substantially, but also lowered sensible heat flux more than the latent heat
flux. With widespread standing water, and little physiological control on
evapotranspiration, the impacts on the partitioning of turbulent fluxes were modest at the
bog compared to the physiologically dominated fluxes at the forested site. Despite the
short duration and singular nature of the event, there was some evidence of a diffuse
radiation fertilization effect when AOD was near or below two. With lighter smoke, both
the wetland and forested site appeared to show enhanced photosynthetic activity (a

greater sink for carbon-dioxide). However, with dense smoke the forested site was a strong carbon source. Given the extensive forest cover in the Pacific Northwest and the growing importance of forest fires in the region, these results suggest that wildfire aerosol during the growing season potentially plays an important role in the regional ecosystem response to smoke and ultimately the carbon budget of the region.

## 1. Introduction

Wildfire activity is projected to increase in frequency and duration over the next century in western North America, primarily as a result of increased summer temperatures, persistent drought and reduced snowpack accompanying climate change (IPCC, 2014; Setelle et al. 2014). In addition to the obvious impacts on visibility and air quality, aerosols arising from biomass burning scatter and absorb solar radiation (direct effect) while also influencing cloud processes by acting as cloud condensation nuclei (indirect effect) (IPCC, 2014). Furthermore, and of particular focus in this work, recent studies point to significant impacts on surface turbulent and radiative fluxes, boundary layer stability and energetics (including cloud development) and carbon exchange between biosphere and the atmosphere (Li et al., 2017).

Radiative impacts of biomass burning are well documented in a variety of settings including South America (Moreira et al., 2017, Sena et al. 2013; Schafer et. al 2002), Africa (Schafer et. al 2002), Spain (Calvo et al. 2010), Russia (Chubarova et al. 2012, Pere et al. 2014), North America (Markowicz et. al. 2017, Vant-Hull et al. 2005) and Asia (Wang et al. 2007). However, there is a growing literature concerned with impacts of smoke plumes on atmospheric boundary layer dynamics as well as surface radiation and energy budgets. For example, Taubman et al. (2004) investigated the impact of a wildfire plume underlain by an urban haze layer in Virginia and Maryland, USA, when aerosol optical depth (AOD) at 500 nm ($\tau_{500}$) varied between $0.42 \pm 0.06$ and $1.53 \pm 0.21$. In that case, atmospheric absorption of solar radiation by the smoke and haze layers resulted in net cooling at the surface and heating of the air aloft, thereby increasing stability. Absorption of solar radiation in the dense smoke layer maintained a morning subsidence inversion and thereby created a positive feedback loop preventing vertical mixing and dilution of the smoke plume itself. Wang and Christopher (2006) report a similar broad

range of impacts on surface radiation/energy budgets, and boundary layer dynamics in a modeling study of the impact of Central American Biomass Burning on source region as well as the Southeastern US (for $\tau_{550}$ =0.09). At the plant canopy scale, Yamosoe et al. (2006) have focused on the impact of biomass burning aerosol on Amazonian forests and have noted an increase in diffuse radiation within the canopy combined with a reduction in total photosynthetically active radiation (PAR) at the top of the canopy. These impacts affected sensible and latent fluxes as well as net ecosystem exchange (NEE) of carbon dioxide ($CO_2$). Subsequently, Steiner et al. (2013) have explored such ecosystem responses using data from six US FLUXNET sites and demonstrate that high AOD reduces midday net radiation by 6%–65% coupled with a 9%–30% decrease in sensible and latent heat fluxes. Niyogi et al. (2004), in an examination of six AmeriFlux sites, conclude that aerosols can exert a significant impact on net $CO_2$ exchange (perhaps more so than clouds) whereby the $CO_2$ sink is increased with aerosol loading for forest and croplands. This effect has become known as the diffuse radiation fertilization effect (DRF) whereby an increase in photosynthesis results from a trade-off between decreased solar radiation and increased light scattering during clouds or smoke (Park et al. 2017, and references therein). It is suggested that the magnitude of the effect is controlled by canopy architecture, leaf area index and plant functional type. Finally, an extensive review of these and other factors (including forest fire aerosols) affecting productivity and carbon fluxes, with a focus on the Northern Australian savanna biome, can be found in Kanniah et al. (2010).

In western Canada, previous studies have examined the chemistry and transport of smoke plumes, as well as the impacts on local air quality (Cottle et al. 2014, McKendry et al. 2011; McKendry et al. 2011). However to date, the impacts on radiation and energy budgets, boundary layer dynamics and ecosystems have not been addressed for biomass burning associated with the coastal temperate coniferous forest biome. As the risk of wildfire increases in such areas (Setelle et al., 2014) biomass burning is likely to have non-trivial impacts on surface climates as well as ecosystem productivity and the carbon cycle in Canada's most productive ecozone.

During the summer of 2015, a rare opportunity arose to investigate such impacts under clear skies and for unusually high AOD ($\tau_{500}$ ~2.5). A period of prolonged drought and elevated temperatures resulted in sustained wildfire activity throughout the Pacific Northwest. In early July 2015, a particularly intense event had significant impacts on air quality and visibility in southwestern British Columbia. Ground level hourly $PM_{2.5}$ (particulate matter less than 2.5 µm in diameter) concentrations approached 200 µg m$^{-3}$ in the city of Vancouver on 5 July and were associated with smoke emanating from approximately 150 km to the north-east in the Pemberton region. (Figure 1). By 7 July reports noted that these Elaho, Boulder Creek and Nahatlatch fires had spread to a combined area of approximately 30,000 ha.

In this study, we focus on the impacts of relatively "fresh" smoke (~1-2 days old) from these intense temperate coniferous forest fires on the radiation budget across four distinct land-use types (a wetland, an urban residential area, an agricultural grass field, and a coniferous forest), as well as surface energy budgets at three of the sites. Finally, we tentatively (given limitations due to the short duration of the event) explore ecosystem response in terms of carbon fluxes at two of the sites (forest and wetland). In so doing we add a new geographic setting to the growing catalog of such ecosystem impacts, and compare the results with studies from other regions. Furthermore, the availability of sunphotometer and aerosol LiDAR data from the immediate area, greatly enriches the information available for interpretation of this event.

## 2. Background and Data Sources
### 2.1 Synoptic Overview

During 2-6 July 2015, western Canada was under the influence of a 500 hPa ridge of high pressure centred off-shore at 135$^{o}$ W. This resulted in northwesterly upper level flow across southwestern British Columbia. At the surface, a thermal trough was located along the western Cordillera, a pattern associated with poor air quality in the region (M$^{c}$Kendry, 1994). Vancouver International Airport recorded maximum daily temperatures in the range 25-27$^{o}$C from 2-6 July with nighttime minima of 21$^{o}$C. Skies were generally cloudless with no precipitation recorded and maximum wind gusts were of ~10 m s$^{-1}$.

Commercial aircraft soundings (Aircraft Meteorological DAta Relay - AMDAR) from Vancouver International Airport (YVR) departures and arrivals show development of a strong surface based inversion late on 4 July that persisted through 5 July 2015, and coinciding with smoke arrival mid-afternoon on 5 July over the western edge of the Lower Fraser Valley (see supplementary material). At that time, the inversion top was at ~500 m above ground and was ~10 K in magnitude. As shown below, this strong inversion effectively trapped the smoke plume below it and was likely responsible for the high particulate matter concentrations and poor visibility observed on 5 July. By 6 July this capping inversion was no longer present and likely disappeared as a result of the evolving weather pattern and advection.

## 2.2 LiDAR

The Environment Canada UBC LiDAR has operated since 2008 at the University of British Columbia (UBC – Totem Field – see Figure 1). This remotely controlled facility was housed in a cargo trailer with modifications including a roof hatch assembly, basic meteorological tower, radar interlock system, climate control system and levelling stabilizers. A Continuum Inlite III (small footprint) laser operating at 1064/532 nm simultaneously with a pulse repetition rate of 10 Hz is the foundation of the system. The upward-pointing system measures the return signal in three channels (1064 nm, and two polarization channels at 532 nm). The system is described in detail by Strawbridge (2013), and an example of its application shown in Cottle et al. (2014).

## 2.3 AERONET/AEROCAN

The global AERONET (AErosol RObotic NETwork) has operated since 1993 and is focused on measurements of vertically integrated aerosol properties using the CIMEL sunphotometer/sky radiometer instrument (Holben et al., 1998). AEROCAN CIMELs (AEROCAN is the Canadian sub-network of AERONET) include a facility on Saturna Island 55 km to the south of the UBC CORAL-net site. Here, solar irradiances are acquired across eight spectral channels (340, 380, 440, 500, 670, 870, 1020 and 1640 nm) that are transformed into three processing levels of aerosol optical depth (AOD); 1.0 – non-cloud screened; 1.5 – cloud screened; and 2.0 – cloud screened and quality assured. McKendry et al. (2011) demonstrated the application of these data to the transport of

California wildfire plumes. In this paper, as in McKendry et al. (2011), the SDA (Spectral Deconvolution Algorithm) was applied to Level 1.0 AOD spectra in order to better delineate the strongly varying contribution of fine mode smoke particles at a reference wavelength of 500 nm. Level 1.0 input AODs were chosen to minimize "false negative" smoke-AOD rejection attributed to the Level 1.5 cloud-screening algorithm.

**2.4 Radiative and Turbulent Flux Data**

The smoke event of July 2015 coincided with a period in which routine long term measurements of surface radiation and turbulent fluxes (sensible and latent heat using the eddy-covariance method) were made at three sites in the region, while at a fourth site only the radiation budget was observed (Table 1). Photographs of the sites a detailed description of the instrumentation, discussion of instrumental inter-comparability, corrections applied, and data manipulations are provided in the supplementary materials. Turbulent fluxes were corrected for spike removal, density fluctuations (Webb et al., 1980), and sensor separation effects. Data processing at all sites were cross-checked against standardized Smart Flux processing algorithms (Licor Inc.).

Buckley Bay (Ca-Ca3) is a flux tower with eddy-covariance and radiation sensors measuring exchange between a coniferous forest stand (Douglas-fir, 27 years old) and the atmosphere. The site is located on the eastern slopes of the Vancouver Island Range, about 150 km to the west of Vancouver. Full descriptions of the site and the instruments can be found in Humphreys et al. (2006) and Chen et al. (2009). Burns Bog (Ca-DBB) is a floating platform with eddy-covariance and radiation instrumentation on an open wetland with mosses, sedges, and a significant faction of standing water. Further details of the site are described in Christen et al. (2016) and Lee et al. (2017). Vancouver-Sunset (Ca-VSu) is an urban observational tower above a residential detached urban neighborhood. Details of the instrumentation can be found in Crawford et al. (2014). Vancouver-UBC is a climate station on the Campus of the University of British Columbia that features a full set of radiation measurements.

**Table 1: Measurements and site characteristics**

| Site | Fluxnet ID | Energy Budget | Radiation Budget | NEE | Coordinates (WGS-84) |
|---|---|---|---|---|---|
| **Buckley Bay** Coniferous Forest | **Ca-Ca3** | • | • | • | 124° 54' 1.44" W 49° 32' 4.63" N |
| **Burns Bog** Wetland | **Ca-DBB** | • | • | • | 122° 59′ 5.60″ W 49° 07′45.59″ N |
| **Vancouver-UBC** Grass | - | | • | | 123°14'56.41"W 49°15'19.50"N |
| **Vancouver-Sunset** Residential Urban | **Ca-VSu** | • | • | | 123° 4' 42.24" W 49° 13' 33.96" N |

Based on descriptions and conventions described in Oke (1987), the surface radiation budget can be defined as:

$$Q^* = K_\downarrow - K_\uparrow + L_\downarrow - L_\uparrow$$

where $Q^*$ is the net all-wave radiation), $K_\downarrow$ is the shortwave irradiance comprising direct and diffuse solar radiation, $K_\uparrow$ is the reflected shortwave radiation, $L_\downarrow$ is the longwave ("thermal") irradiance from the sky and $L_\uparrow$ the longwave radiation emitted and reflected from the surface (all in W m$^{-2}$).

The ratio $K_\uparrow / K_\downarrow$ is the surface albedo ($\alpha$) and is the shortwave reflectance of the surface in the solar band. $K_{ext}$ is the extra-terrestrial solar radiation and represents the flux density of solar radiation falling at the outer edge of atmosphere and is computed based on date, time and latitude at the site. The ratio of $K_\downarrow/K_{ext}$ is a measure of the bulk transmissivity of the atmosphere to shortwave radiation. Photosynthetically Active Radiation (PAR), measured at Burns Bog only, is shortwave radiation in the range 440-670 nm and is typically expressed in terms of photon flux density ($\mu$mol m$^{-2}$ s$^{-1)}$.

Furthermore, the non-radiative partitioning of energy partitioning over a surface can be defined in three dimensions using the surface energy balance (Oke et. al, 2017):

$$Q^* + Q_F = Q_H + Q_E + Q_G + \triangle Q_S + \triangle Q_A$$

where $Q_F$ is the heat released inside a volume due to human activities (anthropogenic heat flux), $Q_H$ is the turbulent (convective) sensible heat flux to the atmosphere, $Q_E$ is the turbulent (convective) latent heat exchange with the atmosphere (including evaporation and transpiration), $Q_G$ is the conductive exchange of energy with the underlying

10 substrate, $\triangle Q_S$ the net heat storage in the entire volume above a surface (e.g. urban fabric or plant canopy) and $\triangle Q_A$ the net energy added to or subtracted from a volume due to advection (all in W m$^{-2}$). In the cases examined here, both $\triangle Q_S$ and $\triangle Q_A$ are deemed negligible due to judicious site selection, while $Q_F$ is only of relevance at the Vancouver-Sunset site where it is of order 20 W m$^{-2}$ (Oke et. al. 2017).

The Bowen ratio is defined as $\beta = Q_H/Q_E$ and is a measure of the partitioning of the turbulent heat fluxes. $\beta$ is dependent on availability of water at the surface as well plant physiology and has important consequences for surface climates by influencing both surface temperature and humidity.

Net ecosystem exchange (NEE in $\mu$mol m$^{-2}$ s$^{-1}$) is used in quantifying the carbon balance of an ecosystem and is ecosystem respiration ($R_e$) minus gross ecosystem photosynthesis (GEP), i.e., NEE = $R_e$ – GEP., NEE is negative when the ecosystem is acting as a $CO_2$ sink, and positive when it is acting as a $CO_2$ source.

### 3. Results

### 3.1. Satellite, Lidar and Sunphotometer Observations

MODIS (Moderate Resolution Imaging Spectrometer) imagery for the period is shown in Figure 2a-d. On 4 July 2015 the region was cloud and smoke free. By 5 July a plume of

30 smoke from the fires in the Elaho valley near Pemberton is evident and extends across the southern and central portion of Vancouver Island (including Buckley Bay, but not the three mainland sites). At this time, a "wall of smoke" extended broadly from northwest to

southeast along the Strait of Georgia and slightly to the west of the city of Vancouver. This smoke moved across the city of Vancouver at approximately 15:00 Pacific Daylight Time (PDT) on 5 July 2015 (photographic evidence is shown in supplementary material). HYSPLIT (Hybrid Single Particle Lagrangian Integrated Trajectory Model) modeling
(see supplementary information) at this time confirmed the source, shape and extent of the plume. By 6 July, the plume had dispersed eastward and was accompanied by cloud to the west of Vancouver Island with a signature consistent with a coastally trapped disturbance (Reason and Dunkley, 1993) or marine "stratus surge" as it is commonly known in the region. However, the region to the east of Vancouver Island remained cloud
free, and remained so during the 6 July when dense smoke was still evident across southwestern British Columbia including all four of the measurement sites.

The impact of smoke on air quality in the vicinity of Vancouver is shown in Figure 3. LiDAR imagery (Figure 3c) shows an elevated layer of smoke over the region at ~2000m
elevation prior to the arrival of a "wall of smoke" at ground level at approximately 15:00 PDT on 5 July (depicted by the vertical dashed line). Ground level smoke remained in a shallow layer until approximately 6:00 PDT on 6 July. Subsequently, smoke continued to persist over the region but was confined to a shallow layer at ~1750m elevation AGL. Smoke again descended toward ground level on 7 July but did not reach the surface.
Consequently, $PM_{10}$ concentrations at Vancouver International Airport (Figure 3a) peaked (reaching 250 μg m$^{-3}$) when smoke was at ground level between 15:00 on 5 July and 6:00 PDT on 6 July. (Note, due to the fact that $PM_{10}$ is measured with a *TEOM* instrument and $PM_{2.5}$ by a *Sharp* instrument at Vancouver International Airport, differences in instrument principles and calibrations means that under elevated fine mode
particulate matter conditions, $PM_{2.5}$ values may approach or marginally exceed measured $PM_{10}$ values, as occurred in this case). Over the entire period there was a modest decrease in daytime maximum and minimum temperatures at Vancouver International Airport (Figure 3a).

Analysis of sunphotometer (Saturna Island), LiDAR (UBC) MODIS, AQUA and CALIOP (Cloud-Aerosol Lidar with Orthogonal Polarization) together reveal a complex three-dimensional structure associated with the smoke event (with layers extending into

the 3-6 km range AGL). The event, as shown in both LiDAR and CALIOP data consisted of multiple layers with the predominately fine mode particle signature of smoke confirmed in the AERONET (Saturna Island) data (Figure 3b). Prior to the arrival of the ground level smoke over Vancouver, fine mode AOD values at Saturna Island were very high (~9 at 11:00 PDT on 5 July) and were (on the basis of detailed analysis of the MODIS and AQUA imagery) likely associated with the higher altitude (2-3 km) smoke plumes. The 15:00 PDT "wall of smoke" mentioned above is seen as a sharp fine mode AOD rise at Saturna following the decay of the strong 11:00 peak of the 2-3 km layer (a rise that started around 12:30 PDT on 5 July : the 2 ½ hour difference being a function of the Saturna to UBC transport time and the time that a significant increase in fine mode AOD could be detected at Saturna).

### 3.2 Impact on Radiation and Energy Budgets

The course of diurnal radiation budget components at each site is shown in Figure 4, while daily averages are listed in Table 2. On both 3 and 4 July, all sites show a smooth diurnal course of radiation components consistent with summer clear sky conditions. On these days, mean daily atmospheric bulk transmissivity ($K_\downarrow/K_{ext}$) was approximately 80% at all sites (Table 2). The most dramatic impact of the smoke plume on radiation components occurred on July 5 at Buckley Bay when the mean daily transmissivity dropped to 40% with a reduction in midday $K_\downarrow$ of 49% (to 475 W m$^{-2}$) compared to midday values on 3 and 4 July (~920 W m$^{-2}$). This is consistent with satellite imagery (Figure 2) which shows the smoke layer persisting over Buckley Bay for the entire day on 5 July. At mainland sites, the late arrival of smoke at approximately 15:00 PDT on 5 July is evident in the late afternoon $K_\downarrow$ but had less impact on daily totals. Instead, at these mainland sites the biggest impact of the smoke occurred on 6 July when daily transmissivities dropped to 52-57% and peak midday $K_\downarrow$ values were reduced by approximately 15-25% compared with those observed on the 3 and 4 July. On this day, LiDAR imagery shows the smoke layer to be at a higher elevation with less intense backscatter than seen late on 5 July (Figure 4a).

**Table 2: Daily averages of radiative, turbulent fluxes, Bowen ratio and carbon dioxide fluxes 3-7 July for each of the four sites**

| | 3 July 2015 | 4 July 2015 | 5 July 2015 | 6 July 2015 | 7 July 2015 |
|---|---|---|---|---|---|
| $K_{ext}$ | MJ m$^{-2}$ day$^{-1}$ | | | | |
| | 37.0 | 36.8 | 36.7 | 36.4 | 36.2 |
| Shortwave $K\downarrow$ | MJ m$^{-2}$ day$^{-1}$ | | | | |
| Burns Bog | 29.5 (78%)* | 29.4 (78%) | 25.3 (69%) | 20.8 (57%) | 23.8 (65%) |
| Van.-Sunset | 28.3 (76%) | 28.1 (76%) | 23.4 (63%) | 20.8 (57%) | 21.6 (60%) |
| Van.- UBC | 28.2 (76%) | 27.8 (76%) | 21.5 (59%) | 18.9 (52%) | 19.7 (54%) |
| Buckley Bay | 29.8 (81%) | 29.9 (81%) | 14.4 (39%) | 21.2 (58%) | 24.4 (67%) |
| Albedo (α) $K\uparrow/K\downarrow$ | | | | | |
| Burns Bog | 0.18 | 0.18 | 0.18 | 0.18 | 0.17 |
| Van.-Sunset | 0.16 | 0.15 | 0.17 | 0.17 | 0.16 |
| Van.-UBC | 0.29 | 0.30 | 0.37 | 0.35 | 0.34 |
| Buckley Bay | 0.13 | 0.13 | 0.17 | 0.15 | 0.14 |
| Sensible Heat $Q_H$ | DAYTIME (W m$^{-2}$) | | | | |
| Burns Bog | 67.3 | 63.4 | 30.5 | 60.0 | 97.6 |
| Van.-Sunset | 250.7 | 247.3 | 136.5 | 130.1 | 176.4 |
| Buckley Bay | 232.8 | 217.5 | 41.4 | 136.9 | 179.3 |
| Latent Heat $Q_E$ | DAYTIME (W m$^{-2}$) | | | | |
| Burns-Bog | 116.8 (β=0.58) | 112.8 (β=0.56) | 99.6 (β=0.31) | 89.8 (β=0.67) | 96.4 (β=1.01) |
| Van.-Sunset | 53.6 (β=4.68) | 57.8 (β=4.28) | 62.7 (β=2.18) | 48.3 (β=2.69) | 39.0 (β=4.52) |
| Buckley Bay | 72.4 (β=3.21) | 70.1 (β=3.10) | 49.1 (β=0.84) | 60.3 (β=2.27) | 53.5 (β=3.35) |
| $CO_2$-NEE (daily mean) | g C m$^{-2}$ day$^{-1}$ | | | | |
| Burns Bog | -1.67 | -1.64 | -2.47 | -3.64 | -4.24 |
| Buckley Bay | 0.16 | 1.26 | 1.13 | -1.35 | -2.31 |

* percentage of $K_{ext}$ (extraterrestrial radiation)

5    Daytime Bowen ratio $\beta = Q_H/Q_E$

With respect to remaining radiation budget components, arrival of smoke at all sites was marked by a reduction in $Q^*$. However, variations in $L_\downarrow$ and $L_\uparrow$ were subtle and point to only modest changes in surface or atmospheric temperatures (Figure 3a). Albedo (Table 2) also increased at the two Vancouver sites as well as the Buckley Bay site with the arrival of smoke and likely is a consequence of the reduction in specular reflection during direct solar irradiance and an increase in diffuse reflection.

Diurnal impacts of the smoke event on atmospheric transmissivity are shown in Figure 5 where a clear non-smoke day (3 July 2015) is directly compared with 6 July 2015. Of note, the impact of the low level smoke is apparent in the significant reductions in transmissivity throughout the day. However, near sunrise and sunset, 3 and 6 July have almost the same irradiance at low sun angle, presumably due to the presence of more diffuse light. Secondly, the impacts are the same across all sites/ecosystems and therefore demonstrate a clear regional signal consistent with the widespread smoke distribution shown in Figure 2c.

The course of sensible ($Q_H$) and latent ($Q_E$) turbulent heat fluxes are shown in Figure 6 and summarized in Table 2. As with $K_\downarrow$ above, the most significant impact on $Q_H$ was at Buckley Bay on 5 July where it decreased to 18% (i.e., 41.4 W m$^{-2}$) of clear sky mean daytime time values. At Vancouver-Sunset (6[th] July) and Burns Bog (5 July), the greatest reductions in $Q_H$ were to 52% and 45% respectively of the daytime values on the clear-sky days preceding the episode. The impacts on $Q_E$ were less than for $Q_H$ at all sites. At Burns Bog, the minimum for $Q_E$ occurred on 6 July whereas at Vancouver Sunset it was on 7 July and at Buckley Bay on 5 July. At all sites, $\beta$ was significantly reduced on 5 July with the greatest reduction at Buckley Bay (from $\beta$=3.21 to 0.84). The latter was the result of the large reduction in $Q_H$ at that site (82%) and the relatively small reduction in $Q_E$ (32%). The switch from high direct radiation on 3 and 4 July to predominately diffuse radiation on 5 July was likely responsible for the marked reduction in $Q_H$ as a consequence of reduced heating of leaves in a highly coupled forest canopy (Brümmer et al 2012). In summary, evapotranspiration was maintained at all sites with the wettest sites

(Burns Bog) showing the lowest response to the smoke plume. However, impacts on $Q_H$ were greatest at the forested site (Buckley Bay).

### 3.3 Impact on $CO_2$/NEE and PAR

Daily values for NEE are shown in Table 2 for two sites (Burns Bog and Buckley Bay) where $CO_2$ fluxes were measured over active vegetation. Throughout the smoke period, Burns Bog remained a net carbon sink, and showed an increasingly negative trend in NEE (stronger sink) over the duration of the smoke episode. On 3 July, under clear sky conditions and pre-arrival of smoke the bog was a net $CO_2$ sink (-1.67 g C m$^{-2}$ day$^{-1}$). This was consistent over the seven previous precipitation free days (mean -1.69 g C m$^{-2}$ day$^{-1}$). The peak radiative impact of the smoke at Burns Bog occurred on 6 July (albeit somewhat lesser in magnitude than occurred at Buckley Bay) and was associated with a daily NEE of -3.64 g C m$^{-2}$ day$^{-1}$ (net sink). This effect was even more pronounced on the following day.

Conversely, at the Buckley Bay forested site, the pre-smoke arrival daily NEE on 3 July showed a $CO_2$ neutral situation (0.16 g C m$^{-2}$ day$^{-1}$). Again, as with the Burns Bog site, this was consistent with the seven previous precipitation-free days (mean NEE of -0.08 g C m$^{-2}$ day$^{-1}$), confirming that during such mid-summer conditions (clear, warm and precipitation-free), the Buckley Bay site at the daily scale is broadly $CO_2$ neutral. On 4 July, prior to smoke arrival, and when the peak reduction on incoming shortwave radiation was felt on the 5$^{th}$ of July (with significantly greater solar attenuation than occurred at the three mainland sites), NEE became more positive (a greater atmospheric source of $CO_2$) and then became a strong net sink on 6 and 7 July (-1.35 and $-$ 2.31 g C m$^{-2}$ day$^{-1}$respectively) when the smoke had started to disperse.

Figure 7 shows the course of the $PAR/K_\downarrow$ ratio before and during the smoke episode at Burns Bog. Under clear sky conditions, PAR is roughly a constant fraction of $K_\downarrow$, and for long-term, and all weather conditions, Tortini et al. (2017) found a value of 1.798±0.026 µmol J$^{-1}$ for Burns Bog. This value is comparable to the mid-day values during the first three smoke-free days (July 2 to 4) of 1,789 µmol J$^{-1}$. However, with the arrival of smoke, ratios were reduced significantly to 1.609 µmol J$^{-1}$. This suggests that during

heavy smoke, there are fewer PAR photons available for photosynthesis per energy received, although this does not say anything about the ratio of direct to diffuse.

**4. Discussion**

Based on the observations described above, the presence of a dense layer of wildfire smoke near the surface resulted in significant perturbation of both the radiation and energy budgets over a range of surface types in southwestern British Columbia in early July 2015. The effect was most strongly felt at the forested Buckley Bay site on Vancouver Island.

The dramatic attenuation of incoming shortwave radiation described in section 3.2 is entirely consistent with published literature for forest fire plumes described elsewhere and for a similar range of $AOT_{500}$. Perhaps the best analog is the 2010 fires in central Russia described by Chubarova et al. (2012) and Péré, et al. (2014). Chubarova et al.

(2012) report a 40% loss of shortwave irradiance at $AOT_{500} = 2.5$ (their Figure 10), a value consistent with the losses of 30-50% across our four sites (Table 2) in daytime mean fluxes in $K_{\downarrow}$. Interestingly, Chubarova et al. (2012) observed much greater losses of ~65% for UV radiation (300–380nm) and ~80% for erythemally-weighted irradiance. For the same events, Péré, et al. (2014) examined the shortwave aerosol direct radiative

forcing and its feedback on air and atmospheric temperature over Moscow. For $\tau_{340}$ in the range 2-4, wildfire aerosol caused a significant reduction of surface shortwave radiation (up to 70–84 $\mathrm{Wm^{-2}}$ in diurnal averages) which is again consistent with the ~100 $\mathrm{Wm^{-2}}$ reduction over background in diurnal averages of $K\downarrow$ at the four British Columbia sites. While the focus of this paper is the analysis of the impact of a dense smoke event on

energy balance and ecosystem C fluxes, clouds are also known to show similar effects (Park et al., 2018). We found that at the Buckley Bay site for the June-August 2016

period, clouds reduced mid-day $K_\downarrow$ by as much as 90% relative to the closest clear-sky day, a much greater reduction than with smoke. However, the effects of clouds on ecosystem C and water fluxes are complicated by the influence of other environmental variables such as wind, temperature, associated precipitation and soil moisture.

Similar agreement is apparent when compared with observed reductions of total solar irradiance by forest fire smoke in the Brazilian Amazon and Zambian Savanna (Schafer et al. 2002) At the four sites examined here, total solar irradiance (Table 2) during the fire event represented 50-70% of background values. This is in broad agreement with Brazilian sites (Alta Floresta and Abracos Hill) in Schafer et al. (2002; Figure 1a) which show a reduction of ~68% for $K\downarrow$ over background values for $\tau_{500}$ =2.5. In their study, the African grassland sites show impacts of similar magnitude at somewhat lower AOT values, a likely consequence of the different fuel type, combustion temperatures and aerosol optical properties of the aerosol generated in such fires.

As with radiation budget components, impacts on turbulent heat fluxes were variable across the four sites with the greatest impact at the forested Buckley Bay site where $\beta$ was reduced significantly on 5 July to 0.84 from values of ~3.2 on the preceding clear days. Again, these results are broadly consistent with prior studies elsewhere showing that the impact of aerosol is to reduce $K\downarrow$ (but perhaps increase diffuse radiation) and hence $Q^*$, $Q_H$ and $Q_E$, with the partitioning of the turbulent fluxes $\beta$ appearing to be ecosystem dependent (Steiner et al, 2103). It is important to note however, that FLUXNET data cited by Steiner et al. (2013) for four forested sites, a grassland site and cropland site represent averages from quite different geographical settings than those considered here, and are for $\tau_{500}$ <1.2, significantly less than $\tau_{500}$ values observed in this case.

With respect to Buckley Bay observations, where canopy effects are most important, Yamasoe et al. (2006) offer perhaps a more germane comparison in the context of the Amazon rainforest and for smoke AOT's of a similar magnitude to those observed at

Buckley Bay on 5 July 2015. In their study, both $Q_H$ and $Q_E$ were observed to decrease along with a decrease in photosynthetically active radiation (PAR) due to aerosol attenuation. In this study, Burns Bog offers an interesting contrast to the forested Buckley Bay site. With widespread standing water, and little physiological control on $Q_E$, the impacts on the partitioning of turbulent fluxes were modest compared to the physiologically dominated fluxes at Buckley Bay. The marked reduction in $Q_H$ compared to $Q_E$ (and resulting drop in $\beta$) at Buckley Bay clearly shows the dominating effect of canopy (stomatal) resistance over the much smaller aerodynamic resistance in this highly-coupled forest ecosystem (McNaughton and Jarvis 1983).

Whilst the impact of this intense short duration event on radiation and turbulent fluxes of sensible and latent heat are clear-cut, the impact on carbon fluxes are less certain. In this case, the short duration, spatial variability in smoke density, and singular nature of the event mitigates against the identification of a clear unambiguous signal. Furthermore, the fact that Buckley Bay was the strongest source of $CO_2$ on 4 July, prior to the arrival of smoke (Table 2) suggests that factors other than smoke aerosol were at play in the observed temporal variability of carbon fluxes. However, this case study offers at least a tentative indication of the potential magnitude of a DRF effect in two quite different ecosystems in the Pacific Northwest. In both cases, the arrival of heavy smoke initiated an apparent ecosystem response. Burns Bog, typically a $CO_2$ sink in clear summer conditions, became an even stronger sink with the arrival of smoke. Buckley Bay forest, generally $CO_2$ neutral in such conditions, became a source with the arrival of heavy smoke, and then returned to being a carbon sink on 6 and 7 July when the smoke had started to disperse. The latter hints that as the radiative impact of the smoke diminished, and AOT dropped below the critical threshold of two noted by Yamasoe et al. (2006) and Park et al. (2017), a delayed DRF effect may have been initiated that promoted photosynthesis within the canopy. Further research at both sites under a wide range of smoke events (both duration and intensity) is required.

These broad patterns seem consistent with previous research in different environments (Yamasoe et al, 2006; Nyogi et al. 2004; Park et al., 2017). Yamasoe et al (2006), in the Amazon basin, show smoke caused an increase in the diffuse fraction of PAR, thereby

enhancing transmission deeper into the canopy, leading to enhanced photosynthetic activity and $CO_2$ uptake for moderate $\tau_{500}$. However, of particular relevance to this study, at high $\tau$ (>2), the magnitude of the $CO_2$ flux and NEE decreased, an effect they ascribed to low PAR values and potentially deleterious impacts of pollutants in the smoke itself.

This effect has also been observed by Park et al. (2017) in the central Siberian taiga. There, when $\tau$ >2, a reduction of PAR and diffuse PAR occurred and the forest became a $CO_2$ source. The observation in this study that not only is PAR reduced in dense smoke, but also the ratio of *PAR/ $K_\downarrow$* is diminished when compared to Tortini et al.'s (2017) typical values, also seems to be an potentially important factor contributing to the overall

ecosystem response, and especially the magnitude of the DRF effect. We intend to explore this issue (spectral impacts of smoke) further in the context of more recent fire events and at Buckley Bay forested site in particular.  It should be noted that the impact of smoke on the radiation budget at Burns Bog was significantly less than occurred at Buckley Bay (see table 2). It is therefore likely that the impact on AOT at this site was

also diminished (and may not have exceeded $\tau$ =2) when compared with Buckley Bay. Our results, and those elsewhere, suggest that the ecosystem response to smoke is dependent on the density of smoke and may well be highly variable spatially and temporally, and by ecosystem type and canopy architecture. Clearly further research is required in western North America to identify the major drivers governing ecosystem

response and also the impacts of longer term exposure to smoke.

Finally, we were unable to quantitatively assess the impact of the smoke layer on atmospheric stability in this case. Elsewhere, it has been shown that dense smoke layers provide a positive feedback mechanism by increasing stability and inhibiting cloud

formation. In the absence of a spatial array of vertical soundings and due to the rapidly evolving synoptic situation (where advection was important) we were unable to quantify the radiative effects on the plume layer itself. From available aircraft AMDAR soundings, it was apparent that the plume was trapped by a strong inversion that preceded the arrival of the plume. Certainly, modeling studies in other settings suggest that similar

smoke layers may be subject to radiative heating rates of ~6 K day$^{-1}$ (Calvo et al. 2010, Feingold et al.  2005, Stone et al. 2008) with significant cooling at the surface, thereby

significantly enhancing stability. We propose that a modeling study would help elucidate the processes at play in this case.

**5. Conclusions**

The wildfire smoke episode of early July 2015 in southwestern British Columbia had a significant impact on air quality, the radiation budget and turbulent fluxes of latent and sensible heat. It also appeared to elicit an ecosystem response with respect to NEE of land ecosystems, although this response depended on the overall concentration and we observed enhancements and reductions. Across the four land-use types monitored, impacts were variable, but consistent with published literature in other settings. The greatest impacts on radiation and energy budgets were observed at the forested site where the role of canopy architecture, and the complex physiological responses to an increase in diffuse radiation were most important. Despite the short duration and singular nature of the event, there was some evidence of a DRF effect when smoke density was lower than or close to the threshold of $\tau=2$. With lighter smoke, both the wetland and forested site appeared to show enhanced photosynthetic activity (a greater carbon sink). However, with dense smoke, and significantly reduced irradiance, the forested site was a strong source. This is consistent with literature suggesting that with dense smoke, within canopy PAR is reduced to a point where reduced photosynthetic activity outweighs the DRF effect and the forest becomes a net carbon source (as at night). Given the extensive forest cover in the Pacific Northwest and the growing importance of forest fires in the region, these results suggest that wildfire aerosol potentially plays an important role in the regional ecosystem response to smoke and ultimately the carbon budget of the region. Due to the short duration of the event described here, we recommend further research, including modelling, to elucidate and generalize the patterns observed in this single case.

**Acknowledgements**

We are grateful to the Natural Sciences and Engineering Research Council of Canada (NSERC) for support to individual researchers and graduate students involved in this work. The Burns Bog flux tower operation was funded by Metro Vancouver through contracts to A. Christen. Selected instrumentation was supported by NSERC and CFI. The Sunset Tower site was funded by NSERC with in-kind support by BC Hydro,

while the University of British Columbia and Environment Canada assisted in various ways to support this research. Eric Leinberger did a wonderful job with the figures while Rick Ketler and Zoran Nesic provided invaluable field and technical support. We are very grateful for the constructive comments provided by three anonymous

reviewers.

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

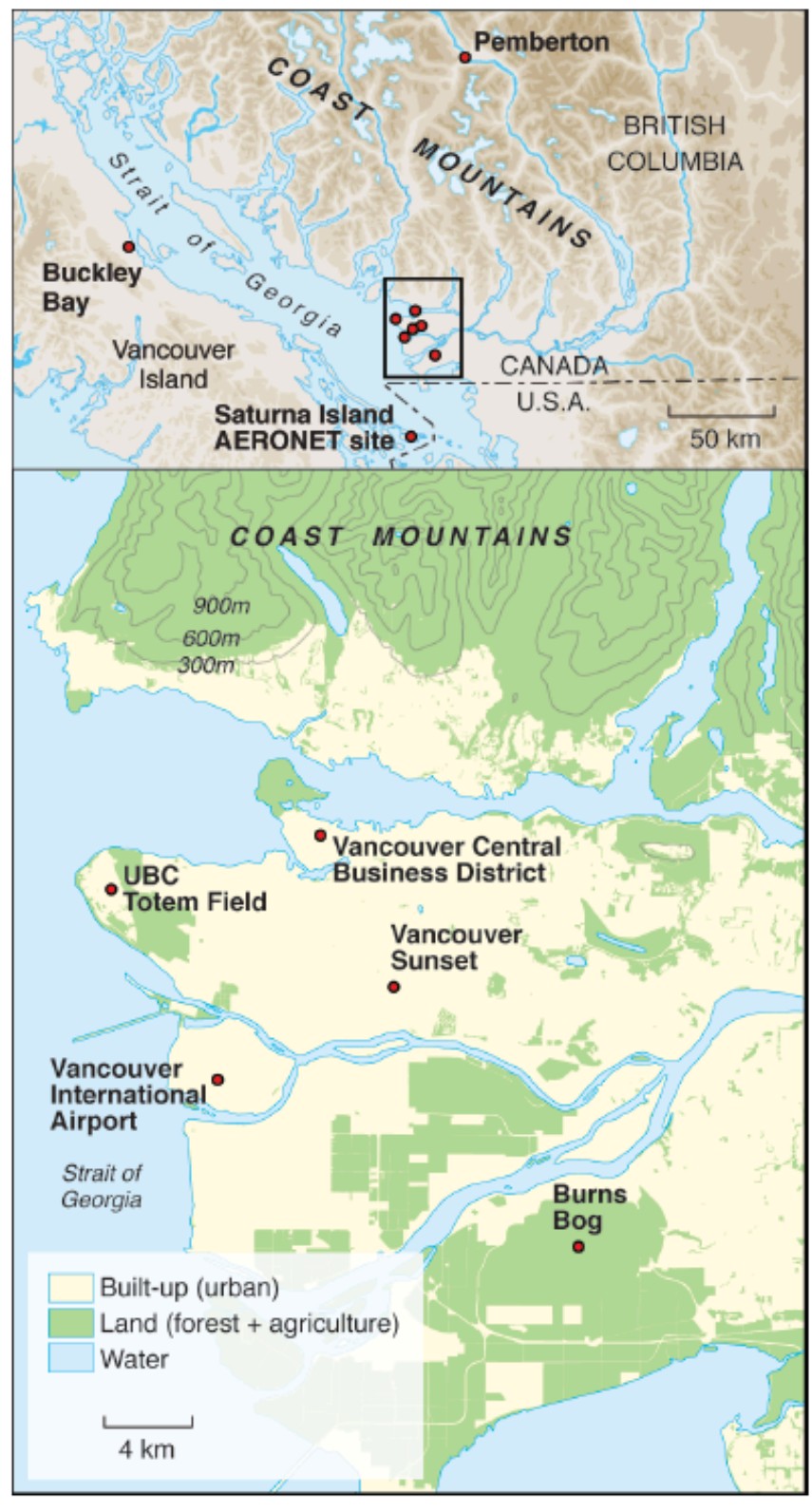

**Figure 1**: Map with inset showing all places mentioned in text

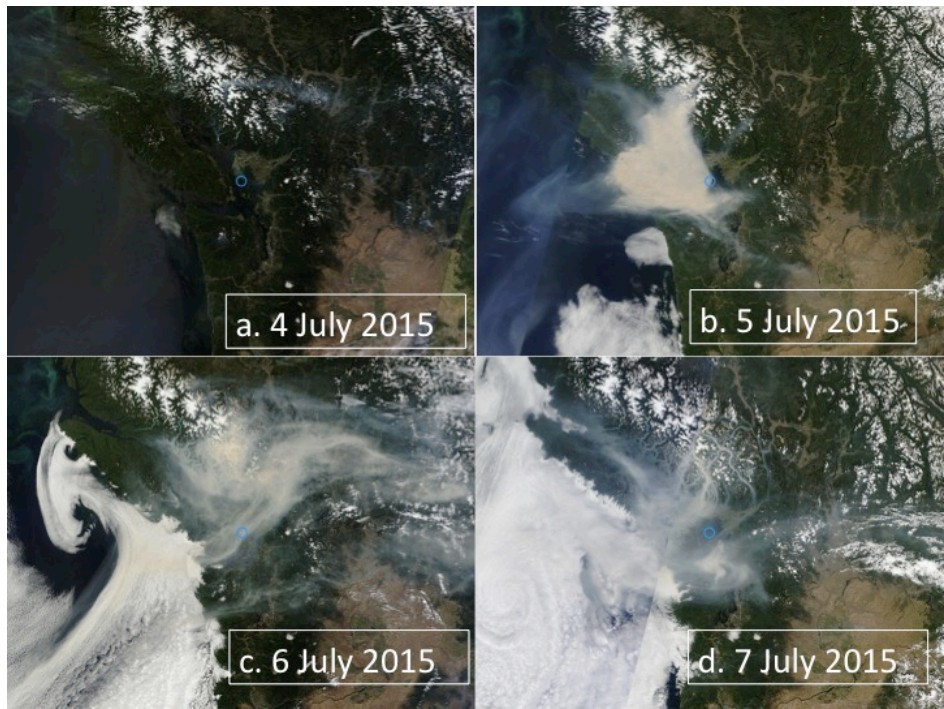

**Figure 2**: Modis Satellite Imagery for the period 4-7 July (Saturna Island shown in Blue circle)

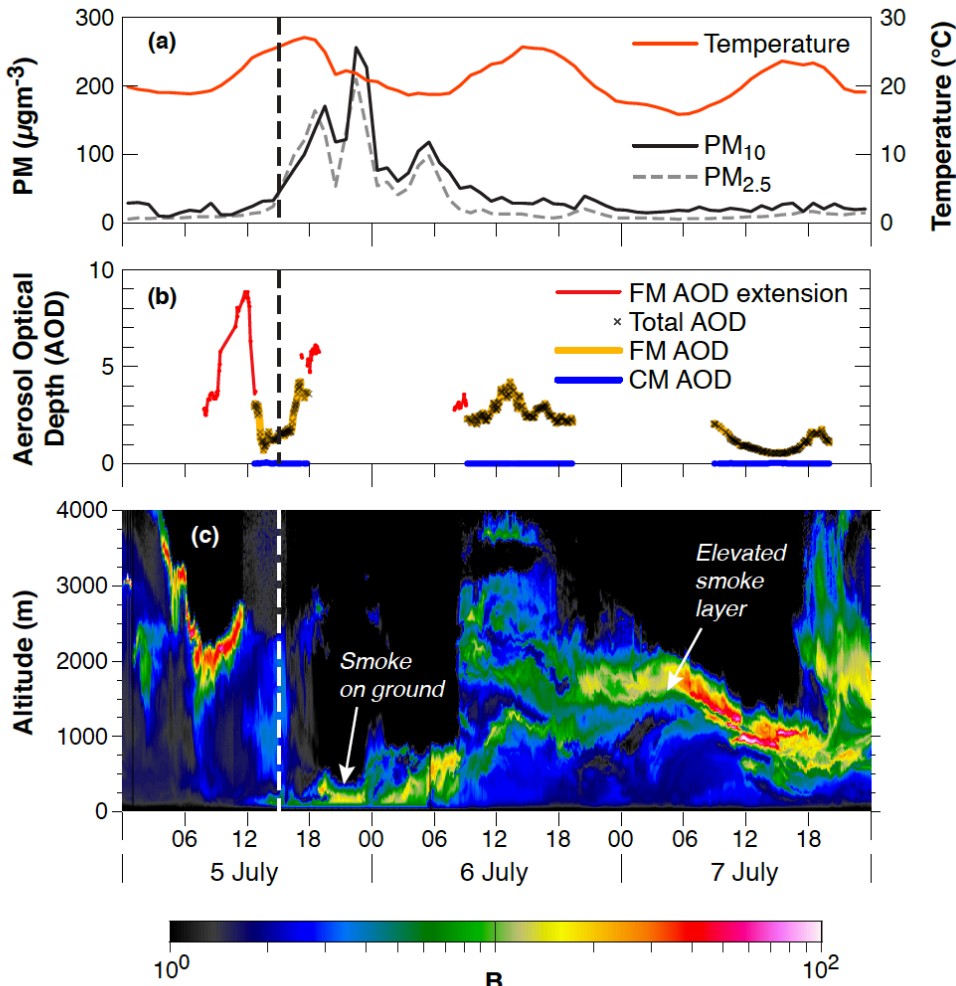

**Figure 3**: Time series for 5-7 July 2017 showing (**a**) $PM_{2.5}$ and $PM_{10}$ observations, and temperature at Vancouver International Airport and (**b**) Aerosol Optical Depths for fine mode (FM) and coarse mode (CM) variation at Saturna Island. The "FM AOD extension" was obtained by assuming that the total AOD at the longer wavelengths of 675 and 870 nm was dominated by the fine mode AOD and extrapolating their AODs back to 500 nm (a choice necessitated by the fact that the extraordinarily large AODs at the shorter wavelengths were eliminated by AERONET processing) and (**c**) LiDAR backscatter from the UBC CORAL-NET site at 532 nm for the period. The red vertical line shows arrival of the low level "wall of smoke" around 3pm on 5 July.

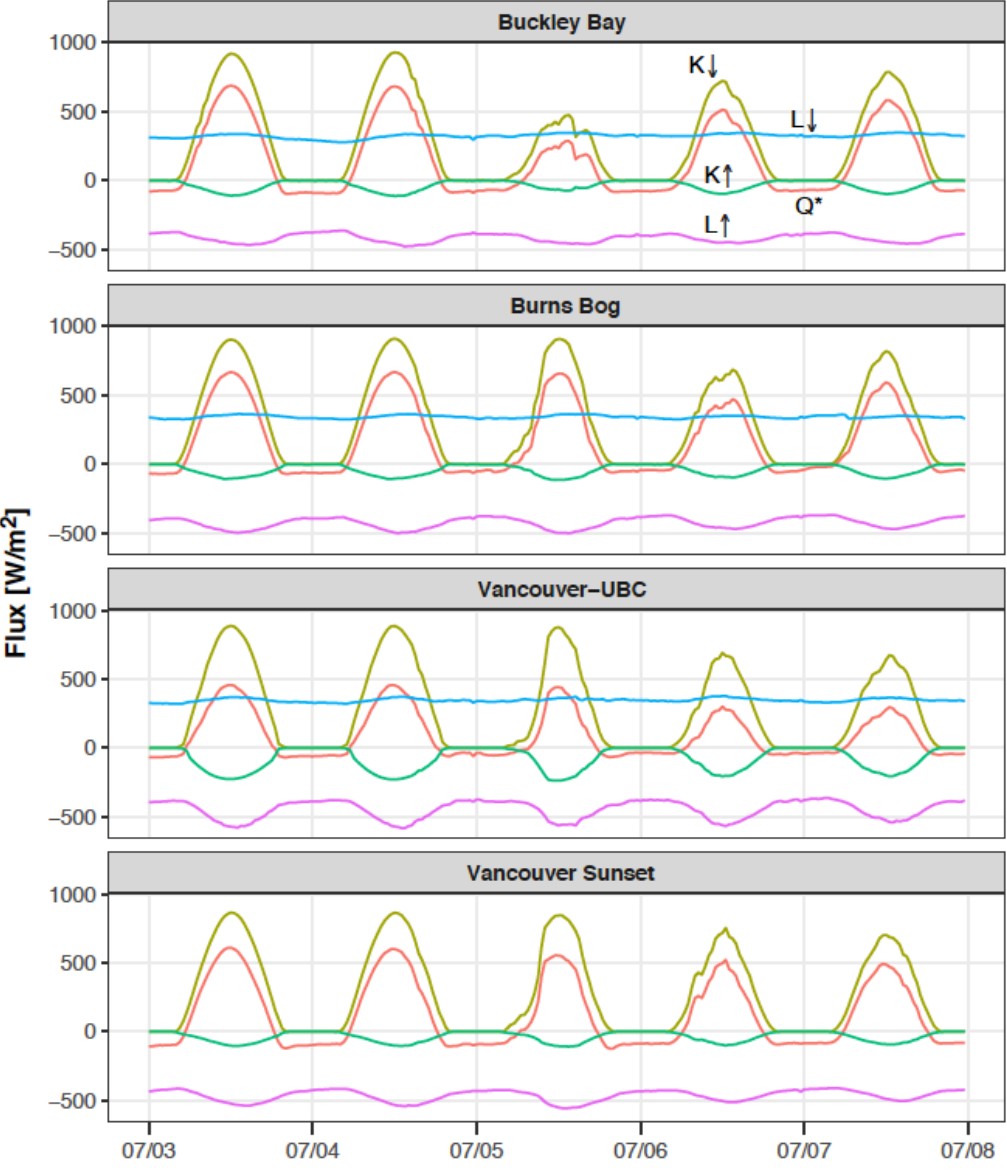

**Figure 4**: Radiation budget components using standard convention. Fluxes away from surface plotted as negative values.

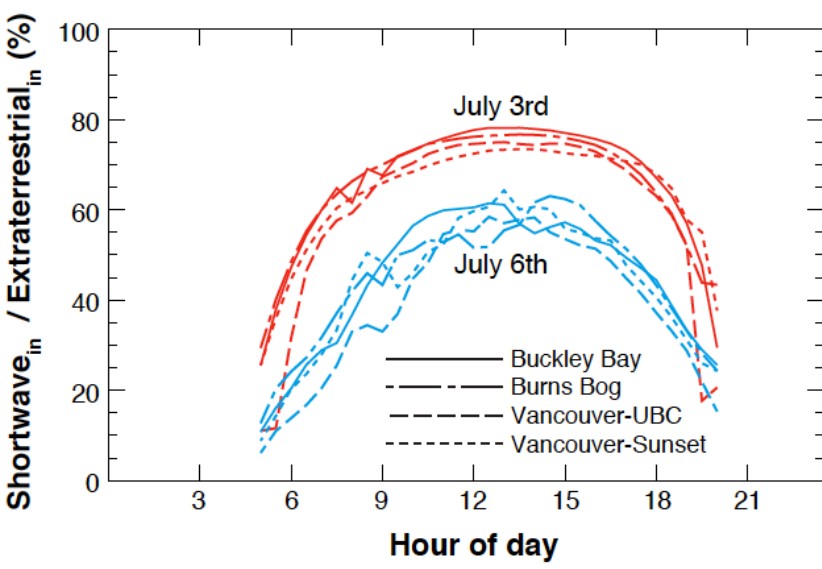

**Figure 5:** Diurnal impacts on incoming solar radiation at each site for 3 (cloudless day) and 6 (smoke) July.

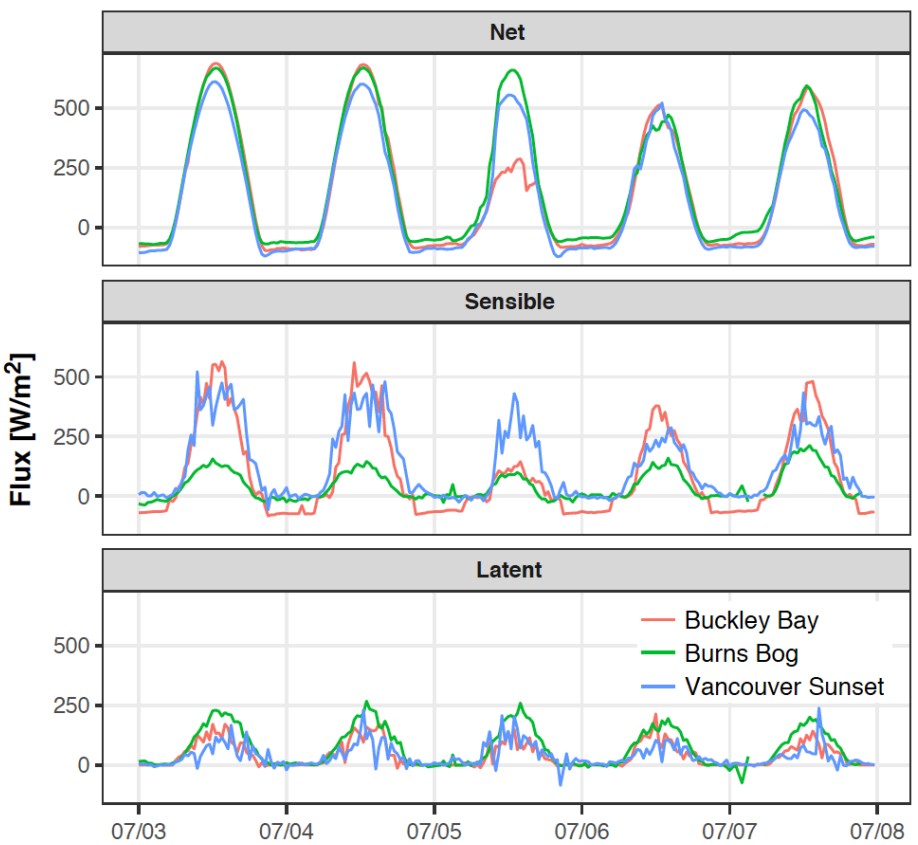

**Figure 6:** Net radiation ($Q^*$), sensible ($Q_H$) and latent ($Q_E$) heat fluxes at three sites. Fluxes away from surface are plotted as positive values.

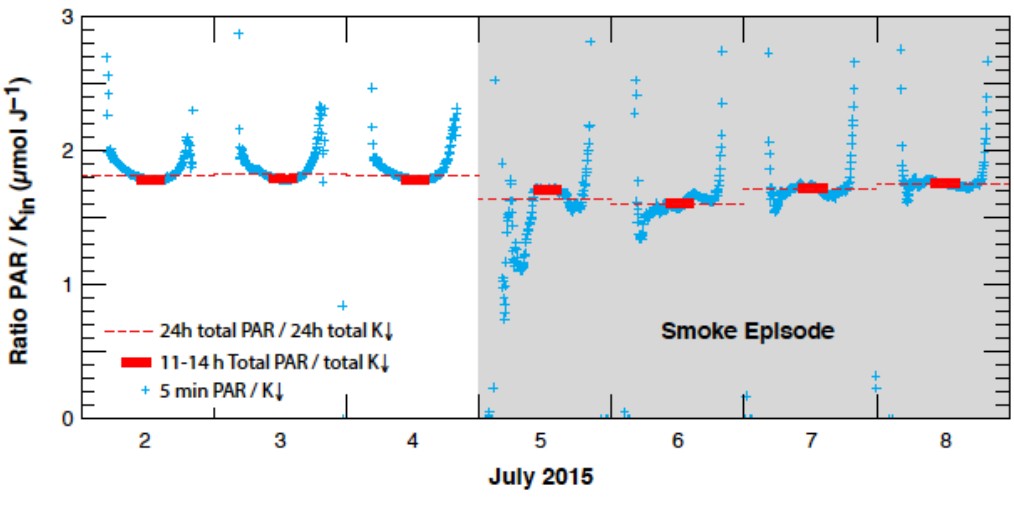

**Figure 7:** *PAR/K*$_\downarrow$ ratio at Burns Bog from 2-8 July 2015

