# Peer review of "Impacts of an Intense Wildfire Smoke Episode on Surface Radiation, Energy and Carbon Fluxes in Southwestern British Columbia, Canada"

_Atmospheric Chemistry and Physics, 2018_

## Referee Comment (RC1) · Anonymous Referee #1 · 12 Jun 2018

The article by McKendry et al. Impacts of an Intense Wildfire Smoke Episode on Surface Radiation, Energy and Carbon Fluxes in Southwestern British Columbia, Canada, which presents results from the study of the impact of a smoke plume on surface radiation transfer, CO2 exchange with the overlying atmosphere and associated convective fluxes is timely and interesting. However, the authors treatment of the energy balance and carbon flux data is superficial and gives no confidence in their reported results. While they refer readers to other articles which discuss each monitoring site, they fail to mention that the instrumentation at each site is different. The authors need to present details of the instrumentation at each site. They need to clearly demonstrate that flux measurements made by different configurations of instrumentation are robust and re-

liable and that reported differences in radiation, trace gas and energy fluxes between sites are real and not the result of different instrumentation.

The authors must report on the corrections made to energy balance data. For example, even in the articles they cite I could not find reference to correction for air density differences (Webb-Pearman-Leuning correction), data spike removal, gap filling (if any), and the influence of solar heating of the LI-7500 on air density within the instrument's measurement path which needs to be corrected. For the urban site – there's no reference to the possible influence of anthropogenic heat flux. I would suggest a much better example of the energy balance equation is presented than the overly simplistic Oke formula.

—————————————————————

---

## Referee Comment (RC2) · Anonymous Referee #2 · 14 Jun 2018

Review of 'Impacts of an Intense Wildfire Smoke on Surface Radiation, Energy and Carbon Fluxes in Southwestern British Columbia, Canada' by McKendry et al.

The study concerns an important problem of wildfire plume effect on solar radiation, energy partitioning and carbon fluxes. Although the topic is interesting and relevant, I doubt the results reported in this study, because the period of smoke is very short (3 days) and the results sound contradictory. Please find the list of comments below.

Specific comments: 1. Average NEE at Burns Bog on the 7th July (plume) almost triples its value as compared to that on the 3rd July (clear sky). The authors state it is due to DRF effect (P. 15, Lines 3-4). However, majority of studies, involving different

[Figure]

ecosystems, report GPP or NEE under diffuse light at most twice that large as it is under low aerosol and clear sky conditions (e.g. review of Kanniah et al., Progress in Phys. Geogr., 2012, 36, 209-237). Also significant leaf area index and canopy height are needed for GPP increase, so it is not clear why diffuse radiation can be an important factor for wetland NEE.

2. AOD is larger than 2 during the whole day on 6 July (Fig. 3). Similarity of incoming radiation at all the sites on that day (Fig. 5) suggests similar AOD for all the sites. However, Buckley Bay and Burns Bog are strong carbon sinks on this day, which is in contradiction with Fig. 7 and the corresponding discussion about other authors' results (P. 15, Lines 18-23).

3. According to Table 2, Buckley Bay was a strong carbon source on 4 July, i.e. before the plume arrived. In fact, it is a stronger source on the 4th July than it was on 5 July under the plume influence. Therefore, it is not clear if this forest became a carbon source on 5 July due to radiation effects (P.12, L. 26-29).

4. About the impact on partitioning of turbulent fluxes: Burns Bog did not experience the same severe radiation conditions as Buckley Bay did on 5 July (Fig. 4). Therefore, one can not conclude that the forest's Bowen ratio is more affected by the plume than the wetland's one (P. 14, L. 25-30). The plume has a moderate influence on the Bowen ratio of the forest ecosystem (similarly to wetland) on all days, except 5 July.

5. The last paragraph in Subsection 3.2 has a lot of contradictions with the numbers reported in Table 2 (the greatest reduction of $Q_H$ at Burns Bog was on 5 July, not on 6 July; at Van Sunset minimum in $Q_E$ was on the 7th July, not on 6 July; beta is reduced only on 5 July at Burns Bog; it is not clear what 90% reduction (?) at Burns Bog refers to).

6. In the abstract, the authors announce analysis for 'four land-use types', but the energy fluxes are reported for three sites and the carbon flux - for two sites. Also in the introduction (p. 4 lines 5-10), the authors mention '...turbulent fluxes and ecosystem

responses of four distinct land-use types' which they do not report. This is misleading, because energy and carbon fluxes are the important components of the study.

7. In the introduction, the energy partitioning problem is not addressed. It is DRF effect that is discussed in the abstract, introduction and conclusion, but there are no figures showing NEE/GPP and diffuse radiation in the manuscript.

---

## Referee Comment (RC3) · Anonymous Referee #3 · 14 Jun 2018

General Comments:

In this study the authors describe the effect of a short-duration wildfire smoke episode on radiation and energy budgets as well as carbon fluxes at differing land surface sites in southwest British Columbia. They they compare their results with those from other geographic settings. The work adds to the literature on the effects of wildfire smoke, by adding a case study in this region. The paper is well written and clearly presented.

My only major suggestion would be that the authors provide some additional context for the smoke-induced changes, by for example comparing them with changes resulting from a cloudy vs a clear day so that a reader could get a sense of the significance of the

smoke-induced changes in relation to "normal" weather-induced changes in radiation, energy and carbon balances.

Minor Comments:

1. P2L26: "…was nearly equal to the attenuation at the surface…" – do you mean nearly equal to the absorption by the surface? 2. P5L26: define SDA 3. P8L31: are PM concentrations referred to here PM10 or PM2.5?; also earlier in the data sources section you should state what instruments are used to measure PM2.5 and PM10 and account for the PM2.5 readings being higher than the PM10 readings on the afternoon of 5 July (as of course PM2.5 is a subset of PM10 so should be <= to it…) 4. P7L13-19: times are given, but clarify in the text that the date is July 5 5. P9L3-4: promises further discussion of the role of smoke on temperature below, but I could not find much further discussion of this…

---

## Author Comment (AC1) · 21 Jun 2018

We thank the three reviewers for their positive and constructive comments. We are currently formulating a detailed response that addresses the issues raised. In particular we are preparing supplementary materials including a large table that provides details on the instrumentation and methods used. We also propose to add further materials and clarifications regarding other points raised. (e.g. a cloudy day for comparison purposes. Note, a clear day example is already provided). The concern raised regarding the short duration of the event is interesting, and one that we grappled with ourselves. On one hand, we feel that in western North America, short events such as the one

described are not uncommon due to mid-latitude synoptic variability. The example examined is therefore somewhat representative and offers the opportunity to explore the rapidity of ecosystem response. That said, significantly longer events would allow more robust conclusions to be drawn. An unusually long (two weeks), but less intense, smoke event in 2017 is currently under investigation for this reason. We welcome more discussion around this issue.

---

## Author Response (AR1)

**Response to Reviewers + marked up manuscript + Supplementary materials**

5 **General Comments:**
We thank the three reviewers for their constructive comments. We have undertaken a significant revision in order to address the various concerns raised. The specific detailed concerns and our responses are shown below in 2 column tabular form. Specifically we have:

10       1. Added material to describe the instrumentation and data analysis to provide evidence that data from each site are indeed comparable (supplementary materials). Several references were added to support this and added to the main reference list.

       2. Changed the description of the energy budget to accommodate anthropogenic,
15       storage and advective heat fluxes as described in the recent text Oke et al. (2017)

       3. Significantly modified our conclusions regarding the carbon flux impacts, which reviewer 2 rightly sees as less convincing than the other impacts. We appreciate the Kanniah et al. (2010) reference which is now cited - it influenced our new
20       interpretation of the results.

       4. Completed other requested clarifications and corrections including replacing site photographs in the supplemental materials.

25 The most significant concern raised, we believe, is from reviewer 2 who states: "I doubt the results reported in this study, because the period of smoke is very short (3 days) and the results sound contradictory". On revisiting the original data, we agree that this a very valid concern, especially with respect to the carbon fluxes. Most importantly, we have changed and enhanced our discussion and conclusions regarding the DRF effect
30 throughout by acknowledging the concerns raised by the reviewer. We also distinguish between the carbon flux results, with their attendant limitations and contradictions, and the more robust radiation and energy balance components of the study.

In writing the manuscript, we were somewhat conflicted as to whether we should add in
35 results from more recent, longer duration events that are currently under investigation (e.g. the event lasting the first two weeks of August 2017 and the recent event of Late July 2018). However we believed ultimately that there is merit in considering this short, singular event as something of a serendipitous natural experiment. For example it permits the opportunity (albeit for a single case) to at least explore the rapidity and magnitude of
40 ecosystem response, and for the first time in our region, under what was an unusually dense pall of smoke. We will continue this research, including integrating a range of cases to develop more robust statistical relations. The reviewer also rightly noted contradictions and consistencies in Table 2. We have reworked this section to better tie together the results and discussion and better reflect these uncertainties.
45

| REVIEWER 1 | RESPONSE |
|---|---|
| The authors need to present details of the instrumentation at each site. They need to clearly demonstrate that flux measurements made by different configurations of instrumentation are robust and reliable and that reported differences in radiation, trace gas and energy fluxes between sites are real and not the result of different instrumentation. | We agree - we have added a table to the supplementary materials and discussion of corrections etc. Given the long and rich history of publications emanating from each site, and the necessity to adhere to international standards and protocols (e.g. Fluxnet) we feel confident in the intercomparability of each data set.

P6 Line 5: added "a detailed description of the instrumentation, discussion of instrumental inter-comparability, corrections applied, and data manipulations are provided in the supplementary materials. Turbulent fluxes were corrected for spike removal, density fluctuations (Webb et al., 1980), and sensor separation effects. Data processing at all sites were cross-checked against standardized Smart Flux processing algorithms (Licor Inc.). |
| The authors must report on the corrections made to energy balance data. For example, even in the articles they cite I could not find reference to correction for air density differences (Webb-Pearman-Leuning correction), data spike removal, gap filling (if any), and the influence of solar heating of the LI-7500 on air density within the instrument's measurement path which needs to be corrected | Corrections are described in the references listed in the supplementary material, however we acknowledge that it is important to list directly the corrections as follows in the main text:

"Turbulent fluxes were corrected for spike removal, density fluctuations (Webb et al., 1980), and sensor separation effects. Although not all sites used the same processing, all data were cross-checked against standardized SmartFlux processing algorithms (Licor Inc.)"

A correction for solar heating of the LI-7500 on air density within the path is not required at the given warm temperatures as we show in the supplement of Cassidy et al. (2015). |
| For the urban site – there's no reference to the possible influence of anthropogenic heat flux. I would suggest a much better example of the energy balance equation is presented than the overly simplistic Oke formula | Agreed: we have replaced the old Oke formulation with the three dimensional energy balance equation that includes anthropogenic heat flux, storage and advection. We have added the citation to the new Oke et al. "Urban Climates" book (2017) and noted the magnitude of the |

| | anthropogenic heat flux at the urban site. |
|---|---|
| | "Furthermore, the non-radiative partitioning of energy partitioning over a surface can be defined in three dimensions using the surface energy balance (Oke et. al, 2017): $$Q^* + Q_F = Q_H + Q_E + Q_G + \triangle Q_S + \triangle Q_A$$ where $Q_F$ is the heat released inside a volume due to human activities (anthropogenic heat flux), $Q_H$ is the turbulent (convective) sensible heat flux to the atmosphere, $Q_E$ is the turbulent (convective) latent heat exchange with the atmosphere (including evaporation and transpiration), $Q_G$ is the conductive exchange of energy with the underlying substrate, $\triangle Q_S$ the net heat storage in the entire volume above a surface (e.g. urban fabric or plant canopy) and $\triangle Q_A$ the net energy added to or subtracted from a volume due to advection (all in W m$^{-2}$). In the cases examined here, both $\triangle Q_S$ *and* $\triangle Q_A$ are deemed negligible due to judicious site selection, while $Q_F$ is only of relevance at the Vancouver-Sunset site where it is of order 20 Wm$^{-2}$ (Oke et. al. 2017)." |
| **REVIEWER 2** | |
| I doubt the results reported in this study, because the period of smoke is very short (3 days) and the results sound contradictory | See extended response above. We agree with respect to Carbon fluxes and have modified accordingly. The radiative and turbulent flux results we believe are robust |
| Average NEE at Burns Bog on the 7th July (plume) almost triples its value as compared to that on the 3rd July (clear sky). The authors state it is due to DRF effect (P. 15, Lines 3-4). However, majority of studies, involving different ecosystems, report GPP or NEE under diffuse light at most twice that large as it is under low aerosol and clear sky conditions (e.g. review of Kanniah et al., Progress in Phys. Geogr., 2012, 36, 209-237). | See extended response above. We have modified text and cited Kanniah, K.D., Beringer, J., Hutley, L.B. : The comparative role of key environmental factors in determining savanna productivity and carbon fluxes: A review, with special reference to northern Australia, Progress in Physical Geography, 34(4), 459-490, 2010. |

| | |
|---|---|
| Also significant leaf area index and canopy height are needed for GPP increase, so it is not clear why diffuse radiation can be an important factor for wetland NEE. | Again, we appreciate and share the reviewers concerns and have modified text to reflect this. |
| AOD is larger than 2 during the whole day on 6 July (Fig. 3). Similarity of incoming radiation at all the sites on that day (Fig. 5) suggests similar AOD for all the sites. However, Buckley Bay and Burns Bog are strong carbon sinks on this day, which is in contradiction with Fig. 7 and the corresponding discussion about other authors' results (P. 15, Lines 18-23). | Thanks – we agree. As stated above, we have significantly modified the abstract, discussion and conclusions to reflect these contradictions.
For example, in discussion we have added

"Whilst the impact of this intense short duration event on radiation and turbulent fluxes of sensible and latent heat are clear-cut, the extent to which there were unambiguous impacts on carbon fluxes is less certain. In this case, the short duration, spatial variability in smoke density, and singular nature of the event mitigates against such a finding. Furthermore, the fact that Buckley Bay was the strongest source of $CO_2$ on 4 July, prior to the arrival of smoke (Table 2) suggests that factors other than smoke aerosol were at play in the observed temporal variability of carbon fluxes. However, this case study offers at least a tentative indication of the potential magnitude of a DRF effect.." |
| According to Table 2, Buckley Bay was a strong carbon source on 4 July, i.e. before the plume arrived. In fact, it is a stronger source on the 4th July than it was on 5 July under the plume influence. Therefore, it is not clear if this forest became a carbon source on 5 July due to radiation effects (P.12, L. 26-29). | We re-examined webcam photos and raw data from the site and agree with the reviewers observation. Consequently we have modified our discussion accordingly. See response above. |
| About the impact on partitioning of turbulent fluxes: Burns Bog did not experience the same severe radiation conditions as Buckley Bay did on 5 July (Fig. 4). Therefore, one can not conclude that the forest's Bowen ratio is more affected by the plume than the wetland's one (P. 14, L. 25-30). The plume has a moderate influence on the Bowen ratio of | We agree and thank the reviewer– the last paragraph of section 3.2 has been significantly reworded to accurately reflect the data presented in Table 2. |

| | |
|---|---|
| the forest ecosystem (similarly to wetland) on all days, except 5 July. 5. The last paragraph in Subsection 3.2 has a lot of contradictions with the numbers reported in Table 2 (the greatest reduction of Q_H at Burns Bog was on 5 July, not on 6 July; at Van Sunset minimum in Q_E was on the 7th July, not on 6 July; beta is reduced only on 5 July at Burns Bog; it is not clear what 90% reduction (?) at Burns Bog refers to). | |
| In the abstract, the authors announce analysis for 'four land-use types', but the energy fluxes are reported for three sites and the carbon flux - for two sites. Also in the introduction (p. 4 lines 5-10), the authors mention ' turbulent fluxes and ecosystem responses of four distinct land-use types' which they do not report. This is misleading, because energy and carbon fluxes are the important components of the study. | We agree – the abstract has rewritten to reflect the different measurements made at different sites. |
| In the introduction, the energy partitioning problem is not addressed. It is DRF effect that is discussed in the abstract, introduction and conclusion, but there are no figures showing NEE/GPP and diffuse radiation in the manuscript. | If we correctly understand the question, you mean the partitioning between sensible and latent heat flux.

 In fact we do discuss the energy budget partitioning in the abstract as "the impacts on the partitioning of turbulent fluxes were modest" and the introduction as "These impacts affected sensible and latent fluxes as well as net ecosystem exchange (NEE) of carbon dioxide ($CO_2$). Subsequently, Steiner et al. (2013) have explored such ecosystem responses using data from six US FLUXNET sites and demonstrate that high AOD reduces midday net radiation by 6%–65% coupled with a 9%–30% decrease in sensible and latent heat fluxes."

 We added the following sentence in the abstract:

 "At the forest site, the arrival of smoke reduced both sensible and latent heat flux substantially, but also lowered sensible |

| | heat flux more than the latent heat flux." |
|---|---|
| **REVIEWER 3** | |
| My only major suggestion would be that the authors provide some additional context for the smoke-induced changes, by for example comparing them with changes resulting from a cloudy vs a clear day so that a reader could get a sense of the significance of the smoke-induced changes in relation to "normal" weather-induced changes in radiation, energy and carbon balances. | Agreed: Whilst we have provided the fluxes for a clear smokeless day in advance of the event the provision of data for a "representative" cloudy day, in all their variety, antecedent conditions etc. is more challenging. That said a section is added to discussion to give the sense that cloudy days produce greater reductions in radiation components etc, Section 3.2 second paragraph.

"Whilst the impact of this intense short duration event on radiation and turbulent fluxes of sensible and latent heat are clear-cut, the impact on carbon fluxes are less certain. In this case, the short duration, spatial variability in smoke density, and singular nature of the event mitigates against the identification of a clear unambiguous signal. Furthermore, the fact that Buckley Bay was the strongest source of $CO_2$ on 4 July, prior to the arrival of smoke (Table 2) suggests that factors other than smoke aerosol were at play in the observed temporal variability of carbon fluxes. However, …" |
| P2 L26: "was nearly equal to the attenuation at the surface " – do you mean nearly equal to the absorption by the surface? | We checked the Taubmann paper and they use the term "attenuation at the surface". I agree it is confusing and so given that, we have removed the reference to attenuation at the surface as it adds little to the understanding of the process (absorption by smoke) |
| P5 L26: define SDA | Done – Spectral Deconvolution Algorithm" defined in text |
| P8 L31: are PM concentrations referred to here PM10 or PM2.5?;
also earlier in the data sources section you should state what instruments are used to measure PM2.5 and PM10 and account for the PM2.5 readings being higher than the PM10 readings on the afternoonof 5 July (as of course PM2.5 is a subset of PM10 so | Thanks – we meant $PM_{10}$ – have added subscript in text

Thanks and good point – we have revised the text accordingly.
PM10 is measured using a TEOM but a Sharp is used for PM2.5. Due to differences in the instruments, the |

| | |
|---|---|
| should be <= to it ) | technicians inform me that when the fine mode dominates as in smoke the PM2.5 values can exceed those for PM10 due to the differences in the two instruments.

P9: added:" (Note, due to the fact that $PM_{10}$ is measured with a *TEOM* instrument and $PM_{2.5}$ by a *Sharp* instrument at Vancouver International Airport, differences in instrument principles and calibrations means that under elevated fine mode particulate matter conditions, $PM_{2.5}$ values may approach or marginally exceed measured $PM_{10}$ values, as occurred in this case). " |
| P7 L13- 19: times are given, but clarify in the text that the date is July 5 5.

P9L3-4: promises further discussion of the role of smoke on temperature below, but I could not find much further discussion of this | I believe the reviewer is referring to page 9. Agreed – dates are added to times.

Agreed – we have removed the sentence regarding further discussion as we were not able to discuss impacts of the smoke on stability, nor the effects of advection on temperatures. We hope to address this issue in a future study. |

[revised manuscript text omitted]

**2. Instrumentation and Data Processing**

| Measurement | Site | Instrument | Model | Manufacturer | Height above ground (m) |
|---|---|---|---|---|---|
| **Shortwave Irradiance** | Buckley Bay | 4-Component Net Radiometer | CNR1 | Kipp and Zonen, Delft, The Netherlands | 15 |
| | Burns Bog | 4-Component Net Radiometer | CNR1 | Kipp and Zonen, Delft, The Netherlands | 4.25 |
| | Vancouver-UBC | Pyranometer | CM5 | Kipp and Zonen, Delft, The Netherlands | 1.5 |
| | Vancouver-Sunset | 4-Component Net Radiometer | CNR1 | Kipp and Zonen, Delft, The Netherlands | 26.2 |
| **Reflected Shortwave radiation** | Buckley Bay | 4-Component Net Radiometer | CNR1 | Kipp and Zonen, Delft, The Netherlands | 15 |
| | Burns Bog | 4-Component Net Radiometer | CNR1 | Kipp and Zonen, Delft, The Netherlands | 4.25 |
| | Vancouver-UBC | Pyranometer | CM5 | Kipp and Zonen, Delft, The Netherlands | 1.5 |
| | Vancouver-Sunset | 4-Component Net Radiometer | CNR1 | Kipp and Zonen, Delft, The Netherlands | 26.2 |
| **Longwave irradiance** | Buckley Bay | 4-Component Net Radiometer | CNR1 | Kipp and Zonen Delft, The Netherlands | 15 |
| | Burns Bog | 4-Component Net Radiometer | CNR1 | Kipp and Zonen Delft, The Netherlands | 4.25 |
| | Vancouver-UBC | Pyrgeometer | PIR | Eppley Laboratory Inc., Newport, USA | 1.5 |
| | Vancouver-Sunset | 4-Component Net Radiometer | CNR1 | Kipp and Zonen Delft, The Netherlands | 26.2 |
| **Emitted longwave radiation** | Buckley Bay | 4-Component Net Radiometer | CNR1 | Kipp and Zonen Delft, The Netherlands | 15 |
| | Burns Bog | 4-Component Net Radiometer | CNR1 | Kipp and Zonen Delft, The Netherlands | 4.25 |
| | Vancouver-UBC | Pyrgeometer | PIR | Eppley Laboratory Inc., Newport, USA | 1.5 |
| | Vancouver-Sunset | 4-Component Net Radiometer | CNR1 | Kipp and Zonen, | 26.2 |
| **PAR** | Burns Bog | Quantum sensor | LI-190 | LI-COR Inc., Lincoln, NE, USA | 4.25 |
| **Wind components and sensible heat flux** | Buckley Bay | Ultrasonic anemometer–thermometer | R3 | Gill instruments Ltd., Lymington, UK | 15 |
| | Burns Bog | Ultrasonic anemometer–thermometer | CSAT-3 | Campbell Scientific Inc. (CSI), Logan, UT, USA | 1.8 |
| | Vancouver-Sunset | Ultrasonic anemometer–thermometer | CSAT-3 | Campbell Scientific Inc. (CSI), Logan, UT, USA | 28.8 |
| **$CO_2$ concentration and fluxes** | Buckley Bay | Infared gas analyzer | LI-7200 | LI-COR Inc., Lincoln, NE, USA | 15 |
| | Burns Bog | Infared gas analyzer | LI-7500 | LI-COR Inc., Lincoln, NE, USA | 1.8 |
| **$H_2O$ concentration and fluxes** | Buckley Bay | Infared gas analyzer | LI-7200 | LI-COR Inc., Lincoln, NE, USA | 15 |

| | Burns Bog | Infared gas analyzer | LI-7500 | LI-COR Inc., Lincoln, NE, USA | 1.8 |

| | | |
|---|---|---|
| **Flux data processing and corrections** | Buckley Bay | Matlab, Fluxnet Canada (Morgenstern et. al. (2004), Humphreys et. al. (2006). |
| | Burns Bog | SmartFlux, Lee at al. (2017) |
| | Vancouver-Sunset | Crawford et al. (2012) |

[Figure]

Figure S2: arrival of smoke on 5 July looking north across Downtown Vancouver compared to clear day (Photo courtesy from Elie Bou-Zeid, Princeton University)

**3. AMDAR:**

[Figure]

**Figure S3 – vertical profiles from AMDAR**

**4. HYSPLIT Modelling of the Event**

Dispersion was modeled in order to confirm the smoke source using the NOAA Air Resources Laboratory HYSPLIT (HY-brid Single-Particle Lagrangian Integrated Trajectory) model Version 4 (https://ready.arl.noaa.gov/HYSPLIT.php). HYSPLIT 4 is the current version of a complete system for computing simple air parcel trajectories to complex dispersion and deposition simulations for any location and date (depending on data availability) using a variety of standard data input products (e.g. the NCEP Reanalysis 1948–present). For this case, the concentration fields were calculated for 24 hours for a fire source located at Elaho, covering 10000 ha and started at 0000 05 July 2015 UTC. Concentrations were averaged through a 1500m layer AGL with meteorology driven by the EDAS40 dataset.

[Figure]

Figure S4: Modelled and observed 5 July: HYSPLIT run – 10,000 Ha, EDAS, 1500 m averaged 24 hour started 0000 5 July, UTC